# Influence of Topography and Composition of Commercial Titanium Dental Implants on Cell Adhesion of Human Gingiva-Derived Mesenchymal Stem Cells: An In Vitro Study

**DOI:** 10.3390/ijms242316686

**Published:** 2023-11-24

**Authors:** Vanessa Campos-Bijit, Nicolás Cohn Inostroza, Rocío Orellana, Alejandro Rivera, Alfredo Von Marttens, Cristian Cortez, Cristian Covarrubias

**Affiliations:** 1Laboratory of Nanobiomaterials, Research Institute of Dental Sciences, Faculty of Dentistry, Universidad de Chile, Santiago 8380544, Chile; vcamposb@odontologia.uchile.cl (V.C.-B.); ncohn@odontologia.uchile.cl (N.C.I.); rorellana@odontologia.uchile.cl (R.O.); 2Laboratory of Periodontal Biology, Faculty of Dentistry, Universidad de Chile, Santiago 8380492, Chile; 3Department of Oral and Maxillofacial Surgery, Faculty of Dentistry, Universidad de los Andes, Santiago 8150513, Chile; aleriverapalacios@gmail.com; 4Department of Prosthesis, Faculty of Dentistry, Universidad de Chile, Santiago 8380544, Chile; avonmarttens@gmail.com; 5Escuela de Tecnología Médica, Facultad de Ciencias, Pontificia Universidad Católica de Valparaíso, Valparaíso 2373223, Chile

**Keywords:** dental implant surface, topography, cell adhesion, protein adsorption, retromolar gingiva-derived mesenchymal stem cells

## Abstract

The topography and composition of dental implant surfaces directly impact mesenchymal cell adhesion, proliferation, and differentiation, crucial aspects of achieving osseointegration. However, cell adhesion to biomaterials is considered a key step that drives cell proliferation and differentiation. The aim of this study was to characterize characterize the topography and composition of commercial titanium dental implants manufactured with different surface treatments (two sandblasted/acid-etched (SLA) (INNO Implants, Busan, Republic of Korea; BioHorizons^TM^, Oceanside, CA, USA) and two calcium phosphate (CaP) treated (Biounite^®^, Berazategui, Argentina; Zimmer Biomet, Inc., Warsaw, IN, USA)) and to investigate their influence on the process of cell adhesion in vitro. A smooth surface implant (Zimmer Biomet, Inc.) was used as a control. For that, high-resolution methodologies such as scanning electron microscopy (SEM), X-ray dispersive spectroscopy (EDX), laser scanning confocal microscopy (LSCM), and atomic force microscopy (AFM) were employed. Protein adsorption and retromolar gingival mesenchymal stem cells (GMSCs) adhesion to the implant surfaces were evaluated after 48 h. The adherent cells were examined by SEM and LSCM for morphologic and quantitative analyses. ANOVA and Tukey tests (*α* = 0.05) were employed to determine statistical significance. SEM revealed that INNO, BioHorizons^TM^, and Zimmer implants have an irregular surface, whereas Biounite^®^ has a regular topography consisting of an ordered pattern. EDX confirmed a calcium and phosphate layer on the Biounite^®^ and Zimmer surfaces, and AFM exhibited different roughness parameters. Protein adsorption and cell adhesion were detected on all the implant surfaces studied. However, the Biounite^®^ implant with CaP and regular topography showed the highest protein adsorption capacity and density of adherent GMSCs. Although the Zimmer implant also had a CaP treatment, protein and cell adhesion levels were lower than those observed with Biounite^®^. Our findings indicated that the surface regularity of the implants is a more determinant factor in the cell adhesion process than the CaP treatment. A regular, nanostructured, hydrophilic, and moderately rough topography generates a higher protein adsorption capacity and thus promotes more efficient cell adhesion.

## 1. Introduction

Worldwide, tooth loss is primarily due to untreated caries in permanent teeth and periodontal disease, leading to severe edentulism with increasing age in both sexes [1]. In recent years, implant dentistry has evolved from an experimental treatment to a highly predictable option for replacing missing teeth with implant-supported prostheses [2,3]. In the dental industry, titanium and its alloys have been demonstrated to be non-toxic and more biocompatible than chromium–cobalt and stainless steel [4,5]. While other materials, such as ceramics or polymers, have been used to manufacture dental implants due to their chemical composition, titanium remains the most widely used material [6,7,8]. Titanium has a relative density of 4.5 g/cm^3^, melts at 1677 °C, and boils at 3277 °C; it also has a low thermal conductivity (22 Wm^−1^ K^−1^). It has a high mechanical strength (tensile strength of 240 MPa) with an elongation at a break of 12%. Its modulus of elasticity is relatively low, similar to that of bone (110,000 N/cm^2^), and it is a reactive metal that spontaneously forms a thin layer of titanium oxide (TiO_2_) about 2–10 μm thick when exposed to the environment [9,10,11]. The direct application of Ti implants in contact with tissues places specific requirements on this class of biomaterials in terms of their physicochemical and biological properties, namely biocompatibility, local and systemic safety for the organism, resistance to the effects of physicochemical factors in the oral cavity, and biophysical neutrality [12]. However, cell adhesion to this type of material is not always strong enough, and new formulations and surface modifications should be developed to enhance cell attachment to Ti implants [13]. From the smooth or machined surface implant used as the first prototype, clinicians and researchers have moved on to dental implants structured with macro, micro, and nano topography [14,15,16]. Several components have been evaluated in the fabrication of dental implants, but currently, the materials considered biocompatible today are Ti cp (commercially pure titanium) and titanium alloys, such as (Ti_6_Al_4_V), Al_2_O_3_-based ceramics, hydroxyapatite, calcium phosphates, and zirconia [17,18,19,20]. Physical treatments, such as ion implantation, cathodic sputtering, laser ablation, photofunctionalization, and plasma spray coating with anodization, are commonly used to elevate the thickness of the TiO_2_ layer and create controllable nanostructures on the surfaces of titanium implants [21,22,23,24,25,26]. The combination of physical and chemical treatments, such as sandblasting with large grit and acid-etching (SLA), has been widely used, as this technique produces surfaces with improved wettability and reduced water contact angle [27,28,29]. Regarding surface coatings, an ideal coating should have minimal toxicity, adequate mechanical integrity, and controlled release kinetics [30]. The combined coatings improve osteointegration and prevent infection [20,27]. The most used functional coatings are bioactive ceramics and materials such as α/β-TCPs (alpha and beta tricalcium phosphates), tetracalcium phosphate [Ca_4_(PO_4_)_2_O], hydroxyapatite (HA) [Ca_10_(PO_4_)_6_(OH)_2_]), or bioinert ceramics as zirconium. Additionally, with the advent of new technologies, growth factors (BMP and VEGF), collagen and adhesive proteins [28], bioactive drugs [29], ions and metals [30], and graphene [31,32], among others, have also been incorporated. It has been well established that the topography and composition of the surfaces of dental implants have a direct effect on the adhesion, proliferation, and differentiation of mesenchymal cells, which are crucial processes for osseointegration [14,33]. The basic requirements for successful osseointegration were outlined by Brånemark et al. [34] in the early 1980s. Since then, numerous attempts have been made to adequately define the concept of osseointegration, a complex influenced by several factors, including the immune and nervous systems [35,36,37,38]. From the host’s perspective, the initial healing phase and subsequent osseointegration depend on the availability of osteogenic cells and their ability to adhere and proliferate on the implant surface [39]. Once the implant is placed, water forms a layer around it within seconds, facilitating the adsorption of proteins and other essential molecules on its surface and promoting the adhesion of cells present in this microenvironment [40,41,42,43]. The adhesion process takes place at specific sites on a biomaterial, where integrins enable the connection between the cell cytoskeleton and the extracellular matrix [44]. Integrins promote adhesion and contribute to the differentiation of mesenchymal stem cells on the implant surface. Microtopography-induced extracellular mechanical signals are converted into intracellular biochemical signals through the process of mechanotransduction, which, in turn, activates multiple signaling pathways that ultimately lead to osteogenic differentiation [16,45,46,47]. The evidence reports that regular microscale surfaces, nanostructured at the nanoscale and treated with surface coatings containing bioactive components, exhibit unique properties that promote cell adhesion [48,49,50,51,52,53]. Nanoscale surface modifications increase the surface area [54], which increases protein adsorption capacity and consequently cell adhesion [21,55]. To date, little information is available on the direct culture of cells on the native surface of commercial dental implants for the evaluation of adhesion phenomena [56,57]. The use of implants has been increasing exponentially over the years, and there are few studies evaluating commercial dental implants with a detailed characterization of their surface microstructure and its association with key events involved in osseointegration, such as the cell adhesion phase. Consequently, employing advanced technologies in evaluating biomaterials and their environmental interactions has become increasingly necessary. These technologies allow for obtaining results that enable understanding biological events and interactions at various scales, facilitating the implementation of translational solutions [58,59]. Despite the wide variety of available surface treatments and well-documented physical modifications, the exact influence of these on cell adhesion has not been fully elucidated. We hypothesized that regular nanostructured implant surfaces combined with a bioactive coating would improve cell adhesion. Therefore, the main objective of this study was to evaluate the influence of topography and composition of commercial titanium dental implants on protein adsorption and cell adhesion of GMSCs.

## 2. Results

### 2.1. GMSC Characterization

Retromolar gingiva mesenchymal stem cells (GMSCs) were extracted, as described in the methodology section. Following site-specific extraction (Figure 1A), the enzymatically treated cells were cultured (Figure 1B) and visualized under an optical microscope after seven days of culture. The images revealed a rhomboidal morphology and the presence of extensions that exhibit a typical fibroblastic appearance (Figure 1C). Furthermore, cell surface characterization via flow cytometry revealed a predominance of the stem cell-specific surface markers CD73, CD90, and CD105 in over 85% of the cell population isolated from the retromolar gingiva (Figure 1D,E). Taken together, the results presented in this figure conclusively indicate that the extraction and purification process of pluripotent retromolar gingiva stem cells (GMSCs) were carried out successfully.

### 2.2. Morphological and Textural Characterization

#### 2.2.1. Macrogeometry and Surface Microstructure

In order to evaluate the morphological characteristics, both at a macrostructural and microstructural level, of the implants analyzed in this study, high-resolution images were acquired using a SEM, as shown in Figure 2. These images revealed variations in the shape, distribution, and number of threads throughout all implants. BioHorizons^TM^ presents threads directed towards the apex and together with INNO and Zimmer have interruptions in their structure with flattened areas or cavities. In the same context, the Biounite^®^ implant is characterized by flattened and thicker threads and a mixed macrostructure, where its coronal area differs in relation to the center and apex (Figure 2; top panel 14× and middle panel 27×). The measurements for the macrogeometric analysis of the samples and the calculation of their geometric surface areas are shown in Figure A1. At higher magnifications (Figure 2; middle panel 200× and bottom panel 2000×), it was possible to highlight the greater regularity of the Biounite^®^ surface, which contrasts with the irregular surfaces and disorganized patterns exhibited by the INNO, BioHorizons^TM^, and Zimmer implants. On the other hand, the control lacked a defined microstructure and exhibited a relatively smooth surface.

#### 2.2.2. Roughness

Topography and roughness parameters of the implant surfaces were also analyzed via AFM microscopy. Significant differences in surface roughness were observed between the flank, top, and valley of the INNO/Zimmer and BioHorizons^TM^/Zimmer implants in terms of *Sa* (Table 1). The control and Zimmer implants exhibited the least roughness at all three locations. The INNO, BioHorizons^TM^, and Biounite^®^ implants all had a significantly higher mean roughness than the control (*** p* < 0.01), with Zimmer being the only implant that did not exhibit significant differences from the control (*p* = 0.06). Qualitative images and quantitative roughness parameters were obtained for each implant (Figure 3). The images display a rough micrometric topography in all the implants studied. Regarding the control surface, its topographic irregularities reached a maximum of 0.29 µm (290 nm), unlike the other implants analyzed, where their topographic irregularities (peaks and valleys) reached dimensions of up to 1.2 µm (1200 nm) (Figure 3B). Table 1 presents the results of the various roughness parameters evaluated, which allowed for the samples to be classified into three categories: smooth surfaces, which correspond to the control (Figure 3A); moderately rough surfaces, such as those of INNO, BioHorizons^TM^, and Biounite^®^ (Figure 3B–D); and minimally rough surfaces, which are characteristic of Zimmer (Figure 3E).

### 2.3. Composition

The chemical composition of the implant surfaces was analyzed using EDX coupled to the SEM microscope (Figure 4). This technique made it possible, first, to characterize the distribution (4A) of the elements and quantify (4B) their content in each of the implants. Our results revealed a diversity in composition among the different implant systems, with titanium being the only element shared by all samples examined. Furthermore, the Zimmer and Biounite^®^ implants revealed the presence of phosphorus (P) and calcium (Ca), confirming the information provided by the manufacturer about incorporating these elements as a bioactive coating. EDX analysis also revealed the presence of aluminum (Al) and vanadium (V) in the Zimmer and BioHorizons^TM^ implants, indicating that these implants are manufactured with a classic alloy widely used by various commercial companies (Ti_6_Al_4_V alloy) instead of pure titanium. In the case of the control implant, only Ti and Al were detected, which suggest that this implant is fabricated using a special Ti–Al alloy. Finally, no other elements were found that suggest impurities or contamination, and only one implant system (Biounite^®^) presented sodium (Na) in its composition.

### 2.4. Protein Adsorption

The protein adsorption process on implants is a precondition for cell adhesion in the microenvironment. It has been widely reported that this process creates an interface between the biomaterial and the integrin receptors on the cells. We have performed a protein adsorption assay using fibrinogen as a model protein to evaluate a critical parameter for successful cell adhesion. For this purpose, all implants were incubated with fibrinogen for 6 h at 37 °C, and the concentration of absorbed protein was quantified using colorimetric methods, as detailed in the methods section and Figure 5A. The results showed significant differences in the adsorption capacity between the different implants studied (Figure 5B). In this context, BioHorizons^TM^ exhibited the highest performance: a significantly higher protein adsorption capacity compared to the control and the other implants in the study. They are followed in performance by Biounite^®^ and Zimmer, which exhibited significant differences compared to the control and the INNO implant (Figure 5B).

### 2.5. Cell Adhesion

#### 2.5.1. Assessment of Cell Adhesion Using Scanning Electron Microscopy

Figure 6 shows the SEM images of implant surfaces following 48 h of incubation in contact with cells. It can be observed that GMSCs successfully adhered to all surfaces. A sufficient number of cells were present on the both the valley and the threads areas of the dental implants (500×).

These results illustrate the morphology and arrangement of the cells against each surface. In all implants, there was intimate adhesion to the surface at 48 h, and cell adhesion elements, such as filopodia and lamellipodia, were observed (4000×). In the BioHorizons^TM^, Biounite^®^, and Zimmer implants, the cells mostly acquired a flattened and homogeneous shape on the surface extending widely, while in the INNO implant we observed a rather rounded phenotype (2000×).

#### 2.5.2. Quantification of Cell Adhesion to Different Implant Surfaces

For the qualitative and quantitative analysis of cell adhesion, the implants were examined via laser scanning confocal microscopy using fluorescent markers and different magnifications. Figure 7 depicts the cellular response to implant surfaces, including distribution, morphology, and density. A greater number of cells stood out in the flank and valley of the implants, being more evident in the INNO, BioHorizons^TM^, and Zimmer implants, where an area with a low cell density was observed (highlighted as a black stripe in the DAPI and actin columns) that coincides with the thread of the implant (Figure 7A). The 3D reconstruction of the Z-stack obtained via confocal microscopy confirmed that the cells are uniformly distributed on the original surface of the implants (Figure 7B), regardless of the differences in the macrogeometry exhibited by the different samples. In general, after 48 h, successful cell adhesion was evident in all implants, a phenomenon that was not observed in the control group. A lower cell density was observed in the latter, which translated into a more elongated cellular phenotype, probably due to the greater space available. In accordance with these observations, the quantitative analyses revealed that after 48 h of incubation, the Biounite^®^ and BioHorizons^TM^ implants had a significantly higher number of adhered cells than the INNO and Zimmer implants, as well as the control group (**** p* < 0.00001 and ** p* < 0.01). Notably, the INNO and Zimmer implants did not display significant differences between each other or in comparison to the control group (Figure 7C).

## 3. Discussion

Dental implantology has significantly advanced in the last 30 years, making it essential to clinical dentistry. Although the global market offers over 2500 types of implants from 100 companies, the system choice lacks clear guidelines, mainly depending on marketing, personal preferences, or limited clinical studies. Several examples, covering about 85% of the global market, are Nobel Biocare with 30%, Straumann with 25%, 3i with 15–20%, and Astra with 12%, followed by BioHorizons^TM^, Zimmer, Dentsply Sirona, and OSSTEM [57,60,61]. The research focuses on clinical parameters, while specialists prioritize prosthetic aspects, such as abutment variability, product availability, and compatibility between brands, and simplicity in the prosthetic steps when rehabilitating the implant [62]. Furthermore, most implants on the market today feature controlled surface morphologies at the micron level, although the events crucial for osseointegration occur at the nanometer scale [63]. In the present study, our central purpose was to evaluate the impact of various surface treatments on protein adsorption and cell adhesion and to perform a detailed analysis and description of the phenotype of gingival mesenchymal stem cells (GMSCs) on commercial implants using high-resolution microscopy techniques. Regarding these parameters, the results of this research revealed that surface topography plays a more determining role than the bioactive coating with calcium phosphate. These findings provide a highly relevant contribution since, until now, an exhaustive comparison between these commercial systems has not been carried out. Furthermore, the implementation of high-resolution three-dimensional methodologies, which permitted the direct quantification of cells on the original surfaces of the implants instead of disks, had not been previously documented (Figure 7). The implants were selected for their different surfaces and coatings, as reported by the manufacturers. Two brands were selected that are highly commercialized worldwide and in Chile (BioHorizons^TM^ and Zimmer), along with two newer and cheaper ones (INNO and Biounite^®^). It was essential to analyze the physicochemical characteristics of these implants before analyzing their biological behavior in vitro; hence, a detailed study of all the surfaces was carried out at the level of microstructure, topography, roughness, and composition. Two in vitro studies were performed to study cell adhesion, analyzing protein adsorption and initial cell behavior through direct contact with the materials under controlled working conditions. Several cytomorphometric variables were evaluated to quantitatively and qualitatively assess the cell response on the surfaces.

### 3.1. Characterization of Implant Topography and Composition

The results obtained reveal the differences in the configuration of the macrostructural elements of each implant, such as threads, valleys, and several angles that provide information regarding the direction of the threads. These differences are established by the implant companies mainly in relation to the type of bone in which the implant is destined, with the aim of guaranteeing primary stability in close relation to the surgical technique employed. In 2021, conical implants occupied the first position in the market and had a sale of more than 65%, since they have been reported to be extremely suitable for small spaces where the roots of neighboring teeth are close [64,65]. At higher magnifications using SEM, it is possible to highlight the greater topographic regularity exhibited by Biounite^®^, which contrasts with the irregular and disorganized surfaces exhibited by the INNO, BioHorizons^TM^, and Zimmer implants. The irregular patterns exhibited by the SLA implants (INNO and BioHorizons^TM^) were related to the treatment received. Biounite^®^ presents a surface treatment using nanotechnology applied to electroplating, in which anodizing is conducted by means of chemical plasma, a characteristic that not only makes it regular at the micrometric level but also at the nanometric level. All the implants analyzed in this study have a microrough surface topography. The literature has suggested that a roughness value between 1 µm and 1.5 µm provides an optimal surface for bone integration [66]. Roughness is a factor influencing cell adhesion, and despite available technology, it remains complex to characterize and understand [33]. According to authors such as Kournetas et al., it is necessary to rethink the way roughness parameters are applied to in vitro studies. The three-dimensional roughness value *Sa* is a limited factor in complex topographies and depends on many other factors, such as the technique used or the experimental conditions [67]. The problem with using area-based parameters is that surfaces with extremely different topographies can have similar *Sa* values. This is especially important on surfaces where multiple types of surface treatments are applied, creating micro-roughness and macro-roughness that contribute to the overall roughness and are therefore not reflected in these parameters in a specific way. It would be useful to establish differentiated ranges of microroughness and macroroughness and thus be able to correlate them individually with protein adsorption and cell adhesion [68]. For this reason, in this study, we proposed to complement the observations with different types of high-resolution microscopy, which together allowed us to obtain the best characterization of the surfaces. Nowadays, pure titanium sandblasted and then acid-etched improves mechanical interlocking, which increases the strength between the bone and the implant [69], so the goal of this treatment is more related to mechanical stability than cell adhesion. Wiskott and Belser suggested that stress transfer is also an important variable in osseointegration. Smooth surfaces may not ensure adequate biological and mechanical bonding with the bone surrounding the implant; therefore, the range of stresses induced via a polished surface will be limited [70]. An important finding of in vitro and in vivo studies focusing on bone responses was that moderately rough titanium surfaces (*Sa* between 1 µm and 2 μm) outperformed smoother or even rougher surfaces in terms of osseointegration [66]. Surfaces with extremely low roughness may hinder cell adhesion. We verified the latter by studying the smooth control versus implants. All samples had different compositions with only one element in common, Ti. According to the manufacturer, the Biounite^®^ and Zimmer implants have an additional surface treatment of calcium phosphate applied with different techniques in their manufacture. The presence of these elements in the EDX confirms this premise. It is worth mentioning the differences that these implants present in the amount of titanium (Ti), phosphorus (P), oxygen (O), and calcium (Ca) elements, which has not been always reported by the manufacturer. According to a number of authors, the concentration of calcium does not appear to have a significant effect on the improvement of osseointegration of the modified surfaces, although it could generate differences in the surface thickness of the bioactive coating and influence the adsorption of proteins and cell adhesion [71,72]. P- and Ca-coated implants have been found to have a significantly higher BIC and a higher bond strength to the surrounding bone compared to commercially available pure titanium oxide implants [58]. Regarding the presence of carbon, a thin layer of carbonaceous material, commonly known as adventitious carbon, is usually identified on the surface of most samples exposed to air [73]. Chemical analysis of the dental implants examined in this study revealed the presence of the element carbon (C) in the range from 0.3% to 4.3%, values much lower than those observed in other studies [74]. The literature has suggested that implants containing trace amounts of carbon could be due to adventitious carbon, possibly associated with improvements in osseointegration. However, this suggestion requires confirmation via nanoscale surface analysis using X-ray photoelectron spectroscopy (EDX) [75]. We chose to employ EDX due to its sensitivity and depth of analysis, while taking into account a number of limitations noted with XPS. Recent studies have highlighted fundamental problems in the reliability of XPS data reported on the surfaces of certain metals [76]. It is important to note that carbon is not considered harmful and can be removed via sputtering. Therefore, it should not be associated with possible contamination during sample handling, thanks to the rigorous and meticulous protocols applied in our study during unpacking and subsequent analyses. However, this point remains controversial, as several authors have argued that unintentional carbon residues could include the plastic used in the manufacture and packaging of dental implants [77]. No elements indicative of any type of impurity or contamination, such as silicon, sulfur, or lead, were found and only one implant system (Biounite^®^) presented sodium (Na) in its composition, an element introduced on the surface during the anodizing process.

### 3.2. Protein Adsorption and Cell Adhesion Assays

It is known that the adsorption capacity of proteins on implants is multifactorial and that cells have specific binding sites for certain adhesive proteins (e.g., fibronectin and vinculin) that improve their activity on the surface of the material [42]. Evidently, surface treatment and roughness improve protein adsorption, although the low percentage of adsorption presented by the INNO implant (6.4 ng/mm^2^) is striking, since if we relate some factors, such as the roughness parameter, to adsorption capacity, the trend is maintained in the BioHorizons^TM^, Biounite^®^, and Zimmer implants. Furthermore, it is plausible to speculate that the hydrophilicity reported by the Biounite^®^ and BioHorizons^TM^ implant manufacturers may also positively impact protein adsorption and cell adhesion. According to Parisi et al. [43], under competitive adsorption conditions, as it occurs in the real biological environment, hydrophilicity promotes the selectivity of certain proteins. As a result of selective adsorption, cells on hydrophilic surfaces exhibit increased adhesion, spreading, and proliferation. As we have observed so far, all implants exhibited significant differences in their configuration and composition, which were directly related to the protein adsorption capacity. Despite its smaller surface area and lack of a bioactive surface, BioHorizons^TM^ performed the best in this assay, with a significantly higher (*p* < 0.00001) protein adsorption capacity compared to the control and the other implants. The significantly lower values corresponded to the INNO implant, which did not exhibit significant differences compared to the control (*p* = 0.75). Biounite^®^ and Zimmer possess the same bioactive surface treatment, but the differences in the amount of calcium and phosphorus elements, hydrophilicity, roughness, and differences in the microgeometric distribution pattern apparently influenced the protein adsorption capacity. According to the results obtained, the SEM and LSCM observations indicated that the GMSCs successfully adhered to all the surfaces after 48 h. The Biounite^®^ implant with regular surface and calcium phosphate treatment exhibited the best performance in terms of cell adhesion (13.9 ± 3.4 cells/mm^2^), followed by the BioHorizons^TM^ implant (10.8 ± 3.6 cells/mm^2^). There were no significant differences (*p* = 0.14) in cell adhesion for the latter. This is the first indication that the calcium phosphate bioceramic coating appears to exert no major effect on cell adhesion. BioHorizons^TM^ has an irregular microrough surface and no bioactive coating, so we believed that this result was due to the fact that both implants are hydrophilic and also had the best results in protein adsorption. The BioHorizons^TM^ implant, despite having an irregular surface, is acid-blasted and nitrogen-etched to protect the implant from air and then immersed in a NaCl solution for storage. The advantages of this type of technology include increased hydrophilicity in biological response performance and cell recruitment, which may explain its good performance in protein adsorption and cell adhesion assays [59]. Biounite^®^, on the other hand, with “BioCap” technology is elaborated with nanotechnology applied to electrodeposition treatment, where anodizing is performed by means of chemical plasma, which offers chemical and, at the same time, morphological modifications of the surfaces in a single step. The coating obtained is composed of a titanium oxide matrix enriched with calcium and phosphorus particles, the main constituents of the bone matrix, and has unique nanotubular porosity. This feature not only makes it regular at the micrometric level but also at the nanometric level, a property that favors and ameliorates cell adhesion. However, if we compare the performance of Biounite^®^ with that of Zimmer, which have significant differences in terms of their surfaces (regular vs. irregular) but are both treated with the same surface coating (CaP), the result in cell adhesion revealed significant differences (*p* < 0.0003). The differences in adhesion between Biounite^®^ and its counterpart, Zimmer, in terms of the surface treatment they exhibit led us to believe that the geometric pattern, the regularity of the surface, and the nanotechnology employed in Biounite^®^ are more determining factors in cell adhesion than the presence of calcium phosphate. All in all, the hypothesis was partially accepted. Our results indicated that the micro- and nano-geometric topography is more determinant for cell adhesion than the bioceramic coating with CaP. Based on this, we consider that the presence of calcium phosphate may influence cell differentiation more than adhesion. The literature has indicated that these components have a biological mechanism of action associated with osteoinductive capacity. This is achieved by activating signaling pathways such as ERK1/2 and Wnt/β-catenin, which lead to activating genes involved in osteogenesis, such as RUNX2. Furthermore, its chemical action lies in its presence on the surface of the implants, which generates rapid ion exchange nucleation and, therefore, contributes to the crystallization and formation of bone apatite. These processes commence approximately two weeks after the placement of the implants [23,24]. As reported by Annunziata et al. [60], two types of pure titanium surfaces subjected to an anodic oxidation process were evaluated in comparison with control surfaces. According to the results, the oxidized surfaces ameliorated the adhesion of mesenchymal cells, as well as their differentiation towards the osteoblastic phenotype. These findings have been corroborated by recent studies focused on ameliorating the mechanical properties of these surfaces [61]. On the other hand, since 2011, the concept of surface wettability of a biomaterial has been strongly studied, which, in combination with other surface properties, such as microtopography and nanotopography, surface energy, charge, and functional groups, determines with an influential, yet unknown, weight the biological cascade of events at the biomaterial–host interface, ranging from protein adsorption to tissue interactions and bacterial film formation [78,79].

### 3.3. Concluding Remarks and Future Outlook

In vitro studies possess the limitation that results cannot be extrapolated to the clinical setting, as the conditions under which they are performed are highly controlled and do not consider factors inherent to cell behavior under in vivo conditions, such as the patient’s own physiology. In addition, the nature of the independent sample study design does not allow for statistical correlation of the factors studied, only their relationship. Nevertheless, it is reasonable to assume that the most efficient implants to adsorb fibrinogen also adsorb the proteins in the culture medium’s fetal bovine serum (FBS), which would improve cell adhesion. On the other hand, despite its limitations, the development of in vitro studies is of great importance for generating scientific evidence and for applying the knowledge gained to the design and development of subsequent in vivo studies. This study has provided valuable information that is not always reported by manufacturers and that could be a useful tool in clinical decision making when selecting an implant. The application of Argon 488 laser reflection via confocal laser scanning microscopy on the implants allowed us to observe the distribution of cells on the original surface (and not on the disks) and to obtain 3D images, information very rarely reported in the literature. This not only allowed us to quantify cell adhesion but also to observe how cells are distributed in different regions, such as valleys and threads, of a dental implant. The surface of a dental implant determines the initial phase of the biological response that initiates the osseointegration process, cell adhesion. Each of the four implant systems used in this study has demonstrated positive clinical results according to the manufacturers. However, no information was available to directly compare these implant systems. Although current research is evaluating new surfaces capable of influencing cellular activity, they have not yet been translated into widely available products. Economic reasons may limit and delay adoption. Our understanding of the surface-related responses of living organisms to biomaterials is fragmentary at best. The fabrication of complex surfaces under multiple scales that include differentiated nanotopographies, wetting, antimicrobial, and stable cleanliness properties remain a technical challenge.

## 4. Materials and Methods

### 4.1. Surface Property Characterization

#### 4.1.1. Studied Implants

Commercially available implants from four implant companies were purchased directly from the supplier: INNO (SLA-SH, 3.7 × 10 mm, Busan, Republic of Korea), BioHorizons^TM^ (Tapered Plus SLA, 3.7 × 10.5 mm, Birmingham, AL, USA), Biounite^®^ (ZD BioCap, 4.1 × 10 mm, Buenos Aires, Argentina), and Zimmer Biomet (Tapered Screw-Vent, 4 × 10 mm, Carlsbad, CA, USA). The sizes of implants from different implant companies are not standard, so for comparison purposes, implants of similar dimensions were selected and based on the measurements of their macrostructural characteristics, and their geometric surface areas were calculated (Figure A1). A Zimmer implant analog, (Zimmer Biomet Inc. Carlsbad, CA, USA) made of titanium with a smooth surface, was used as a negative control.

#### 4.1.2. SEM/EDX Analysis

Surface topography and compositions of implants were analyzed via scanning electron microscopy (SEM), equipped with X-ray dispersive energy elemental microanalysis (EDX) in a JEOL model JSM-IT300LV microscope. The implants were individually mounted in the specimen holder and sterile introduced into the microscope for direct observation under low vacuum conditions, without the application of a gold coating. The surfaces of a total of five implants of each type were analyzed. The representative SEM images were acquired at 14×, 27×, 200×, and 2000× of magnification, with an accelerating voltage of 20 kV. Atomic resolution compositional mapping on the implant surfaces was performed via energy dispersive X-ray spectrometry (EDX) (Aztec EDS, Oxford Instruments) coupled to the SEM microscope. The zonal maps allowed for us to know the distribution of the elements on the surface of the implants. The contents (% wt.) of each element were expressed as an average of at least 10-point measurements.

#### 4.1.3. AFM Surface Roughness Measurement

The roughness of the implants was characterized via atomic force microscopy with CoreAFM equipment (NanoSurf, Liestal, Switzerland) used with a 10 × 10 μm^2^ scanner in contact and dynamic force mode. The cantilever was randomly placed on the individual samples, and three images (thread, valley, and flank) were acquired at a resolution of 10 µm^2^. This technique provided a representative analysis of individual areas per implant to characterize surface morphology and roughness. For each area, arithmetic mean height (*Sa*), root-mean-square roughness (*Sq*), maximum height (*Sp*), maximum valley depth (*Sv*), and maximum peak-to-valley distance (*Sz*) were collected as roughness parameters per area (Table 1). Representative images and graphs of the AFM analysis were processed with Gwyddion (Version 2.95, D. Necas & P. Klapetek, Brno, Czech Republic) image analysis software and exported in TIF format. The roughness plots were constructed by drawing an oblique line through the 2D image defined from the analysis of multiple zones of the samples (flank, valley, and thread), where the flank zone was chosen to obtain the 3D reconstruction.

### 4.2. Biological Assessment

#### 4.2.1. Ethics Statement

The protocol and informed consent form for performing human gingiva retromolar biopsies were accepted by the Scientific Ethics Committee of the Faculty of Dentistry of the University of Chile (Acta 2018/06). The inclusion criteria for obtaining retromolar gingival biopsies were based on selecting healthy patients, both orally and systemically, of both sexes and over 18 years of age. The assays were performed in the Nanobiomaterials Laboratory and in the Cell Biology Laboratory at the Faculty of Dentistry of the University of Chile.

#### 4.2.2. Cell Culture

Gingival-derived stem cells were used to evaluate cell adhesion to dental implants. GMSCs were isolated from retromolar gingival biopsies of healthy individuals with a square area of approximately 3 × 3 mm and a depth of 3 mm. Retromolar samples were transported in Eppendorf tubes containing a solution of Dulbecco’s modified Eagle’s medium (α-MEM; Invitrogen Life Technologies, Carlsbad, CA, USA) with 10% fetal bovine serum (FBS GIBCO), 100 U/mL penicillin and 100 mg/mL streptomycin at 10%, and amphotericin B at 1%. The samples were transported at 4 °C to the Nanobiomaterials Laboratory for subsequent cell isolation. Stem cell isolation was performed using a procedure based on the enzymatic digestion technique. Gum samples were washed several times with a PBS solution supplemented with 1% antibiotics (penicillin/streptomycin). The samples were then incubated with 2 mg/mL Dispase II at 4 °C overnight to ensure complete separation of the epithelial surface. They were then minced on Petri dishes into fragments smaller than 1 mm^3^ (Figure 1B) using scalpel blades and digested for 2 h in a solution of α-MEM medium and 10% SBF with 4 mg/mL collagenase I. Each prepared tube was then diluted in a solution of α-MEM medium and 10% SBF, centrifuged, resuspended, and centrifuged again. Samples were seeded on 35 mm plates with α-MEM, 10% SBF, and 1% antibiotics. The medium was replaced every 2–3 days, and the culture was maintained until confluence was achieved in the plates.

#### 4.2.3. Identification and Characterization of GMSCs via Flow Cytometry

The presence of GMSCs in cell isolates was assessed via flow cytometry. Anti-CD73, anti-CD90, and anti-CD105 monoclonal antibodies conjugate with fluorescein isothiocyanate (FITC), phycoerythrin (PE), and allophycocyanin (APC), respectively, were used. At least 2 × 10^5^ cells per well were plated in Eppendorf tubes in 500 μL of PBS. For fluorescence staining, the cells were incubated with FITC-conjugated anti-CD73, anti-CD90, anti-CD105, PE, and APC monoclonal antibodies for 1 h at 4 °C, diluted in PBS with 2% FBS, washed with PBS and fixed with 2% paraformaldehyde (Invitrogen) for 20 min at 4 °C, and analyzed via cytometry.

#### 4.2.4. The Protein Adsorption Assay

The protein adsorption capacity of the dental implant surfaces was determined using human plasma fibrinogen (Calbiochem) as a model protein. A total of 1.5 mL of buffered solution (pH 7.4; K_2_HPO_4_/KH_2_PO_4_; 100 mM) containing 2 ppm of protein was contacted with each implant vertically placed in Eppendorf tubes. Following 6 h of incubation at 37 °C, the implants were removed from the protein solution and washed with phosphate buffer to remove the nonadherent proteins. Then, the adhered proteins were extracted from the implant surfaces by incubating with 1.5 mL of 2% sodium dodecyl sulfate (SDS) solution for 12 hrs. at 37 °C. The concentration of extracted protein was measured using the colorimetric micro bicinchoninic acid assay kit (Ca15045; Thermo Scientific^TM^, Waltham, MA, USA).

#### 4.2.5. Cell Culture on Dental Implants

In order to assess stem cell adhesion capacity, implants were immersed individually and horizontally in a 12-well culture plate and then maintained for 30 min in contact with DMEM (α-MEM; Invitrogen Life Technologies) culture medium supplemented with 10% FBS (GIBCO), L-glutamine (2 mM), and penicillin/streptomycin. The GMSCs were then dispensed onto the implants at a concentration of 4 × 10^5^ cells/mL. The implants were maintained for 48 h in culture at 37 °C and 5% CO_2_. Finally, the samples were removed with tweezers and washed three times in PBS for 5 min for subsequent analysis.

#### 4.2.6. Analysis of Cell Morphology and Adhesion via Confocal and SEM Microscopy

In order to observe cell adhesion to the implants via SEM, cells were fixed with a 2.5% glutaraldehyde mixture. Following fixation, they were washed with PBS and dehydrated through a graded series of increasing ethanol concentrations. The samples were then dried using a CO_2_ critical point dryer (Autodsamdri 815) in the Electron Microscopy Laboratory at the Faculty of Dentistry. Subsequently, the surface was coated with a 10–15 nm gold layer via ion sputtering in a Denton Desk V metallizer. After preparation, the implants were microscopically examined at the magnifications of 500×, 1000×, 2000×, and 4000× to record details of mesenchymal cell attachment to the surface of each implant. For observation by Leica TCS Sp8 confocal laser scanning fluorescence microscope (Leica Microsystem, Wetzlar, Germany), the implants were fixed with a 4% formaldehyde mixture. Following fixation, they were washed with PBS, and ammonium chloride at a concentration of 50 mM was applied for 30 min to remove the autofluorescence of the fixative. Subsequently, the implants were washed 3 times with PBS for 5 min and incubated with a mixture of reagents: Alexa-Fluor^TM^ 555-phalloidin (Thermos Fisher Scientific, Waltham, MA, USA) to visualize actin and DAPI (4,6-diamidino-2-phenylindole dihydrochloride, Thermo Fisher Scientific, Waltham, MA, USA) to visualize nuclei, both diluted in PBS-gelatin (PGN) plus 0.05% saponin. Finally, they were washed 3 times for 5 min with PBS and analyzed. Each implant was marked with a healing abutment to define an observation side. Three zones per side were defined as the observation area: apical (x), middle (y), and coronal (z). The image of stained cells per defined zone were analyzed using the microscopy image analysis program IMARIS (IMARIS software v.7.4.2, Bitplane AG, Andor Technology, Belfast, UK). Image reflection in XYZ coordinates was performed using the reflection application implemented on the Leica SP8 microscope (Figure 7A) by superimposing the excitation and emission wavelengths of the argon laser (488 nm). The Z-stack image acquired with the 10× and 20× dry objectives were employed for the 3D reconstruction of the implant surface.

#### 4.2.7. Cell Nucleus Detection and Quantification

Detection was performed as previously reported [80,81], with several modifications. The series of Z-stack images acquired via confocal microscopy (dry objective 20×) were processed using IMARIS software (v.7.4.2, Bitplane AG, Andor Technology, Belfast, UK) for three-dimensional (3D) reconstructions and detection of the nucleus in XYZ coordinates. Nucleus detection was performed as follows. Confocal images of GMSCs were stained with DAPI (nucleus) and then processed using IMARIS software (v.7.4.2), which allowed for the construction of isospots from the fluorescence signals. Isospots were constructed based on DAPI detection, thus generating several types of isospots with different sizes. As more than 90% of the isospots were detected and three-dimensionally reconstructed with Imaris measuring 13 µm, we used this measurement to quantify all DAPI-positive isospots. Therefore, this method’s systematic application allowed us to conduct a comparative analysis between the different experimental conditions. Finally, an average of the 3 zones per side of the implant was calculated to allow comparisons between samples (Figure A2).

### 4.3. Data and Statistical Analysis

Microsoft Excel 2016 (Microsoft Corp., Redmond, WA, USA) was employed to collate the data. Statistical graphs and data obtained were analyzed using OriginPro 2021 (OriginLab Corp., Northampton, MA, USA) and were reported as the mean ± SD. The data were verified to have a normal distribution with the Shapiro–Wilk test. Differences between groups were evaluated with the ANOVA statistical test and with the Tukey multiple comparison post-hoc test. All experiments were performed in triplicate and worked with a significance error alpha of 0.05 (*p* < 0.05), 95% CI, and a power of 80% (beta of 0.8).

## 5. Conclusions

Despite the inherent limitations of this study, the results obtained support the conclusion that regular surfaces of dental implants at the micrometric and nanometric levels positively influence cell adhesion. The quantity of adhered cells and the quality of said adhesion were significantly improved compared to irregular surfaces. Furthermore, the findings indicate that the regular and ordered topography of surfaces appears to have a more crucial role in cell adhesion behavior compared to the presence of calcium phosphate. Of note, a regular, moderately rough topography with hydrophilic properties promotes higher levels of protein adsorption, which, in turn, favors more robust cell adhesion. Taken together, the results of this study offer promising avenues for future research and valuable biological data for implant selection decisions.

## Figures and Tables

**Figure 1 ijms-24-16686-f001:**
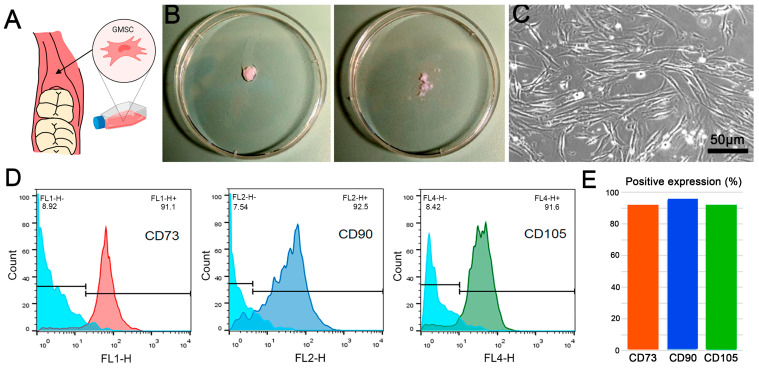
**Isolation and characterization of human gingiva-derived mesenchymal stem cells (GMSCs).** (**A**) Schematic representation of the gingival anatomical site from where the cells in this study were extracted. (**B**) Post-cleaning and post-crushing gingival tissue samples. (**C**) Typical morphology of stem cells isolated from retromolar gingiva obtained via phase-contrast optical microscopy (scale bar: 50 μm, magnification 10×). (**D**,**E**) Expression of markers detected via flow cytometry. Representative histograms and graph showing expression levels of the specific markers CD73, CD90, and CD105 in GMSCs.

**Figure 2 ijms-24-16686-f002:**
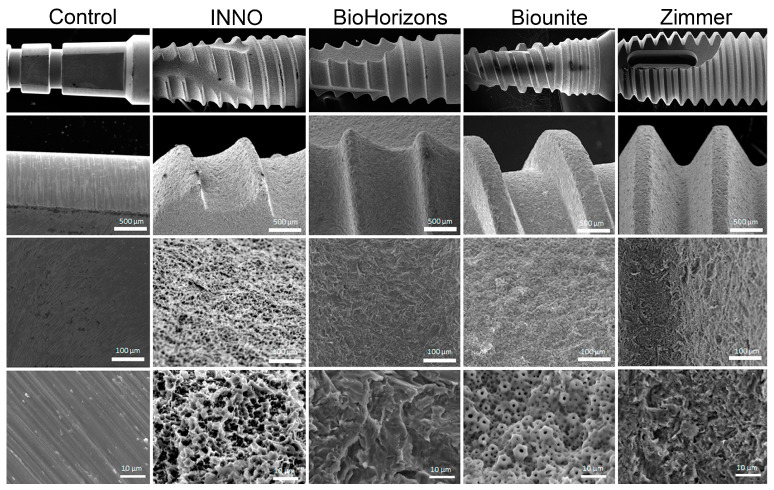
**Surface of the titanium implant observed via scanning electron microscopy (SEM).** Images of the macrostructure and microstructure of the implant surface (scale bar = 500 μm, 100 μm, and 10 μm). In each sample, higher magnification images representative of their microdesign features are observed. The figure exhibits top-to-bottom magnifications of 14×, 27×, 200×, and 2000×, respectively.

**Figure 3 ijms-24-16686-f003:**
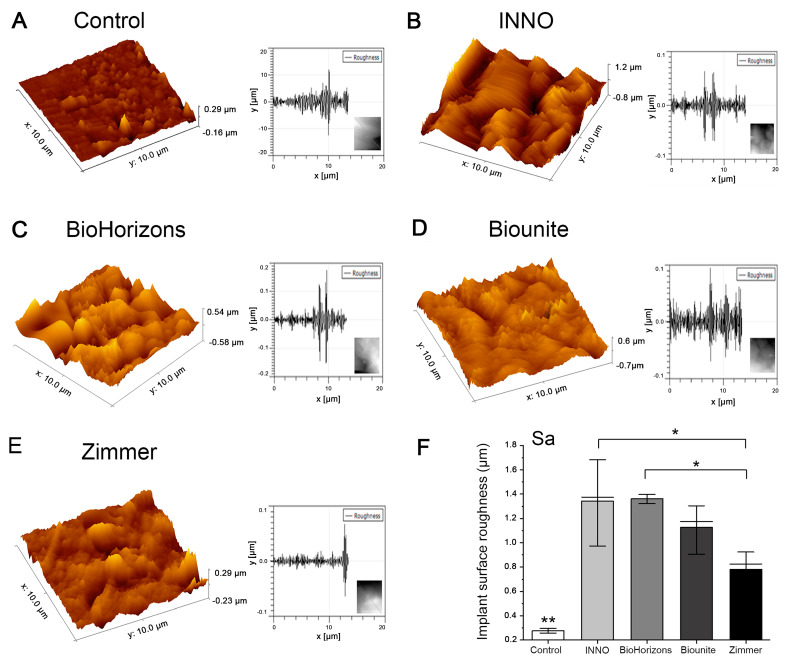
**Images obtained via AFM of the implants.** (**A**–**E**) Three-dimensional (3D) topography images and roughness profile (10 μm^2^). The roughness plots were constructed by drawing an oblique line through the 2D image defined from the analysis of multiple zones of the samples (flank, valley, and top), where the flank zone was chosen to obtain the 3D reconstruction. (**F**) The INNO, BioHorizons^TM^, and Biounite^®^ implants all had a significantly higher mean roughness than the control (** *p* < 0.01), with Zimmer being the only implant that did not exhibit significant differences compared to the control (*p* = 0.06). The only significant differences for the implants were between INNO/Zimmer (* *p* = 0.038) and between BioHorizons^TM^/Zimmer (* *p* = 0.033). Values were analyzed using the OriginPro 2021 statistical program with ANOVA followed by the Tukey post-hoc test, * *p* < 0.05 (*n* = 3 ± SD).

**Figure 4 ijms-24-16686-f004:**
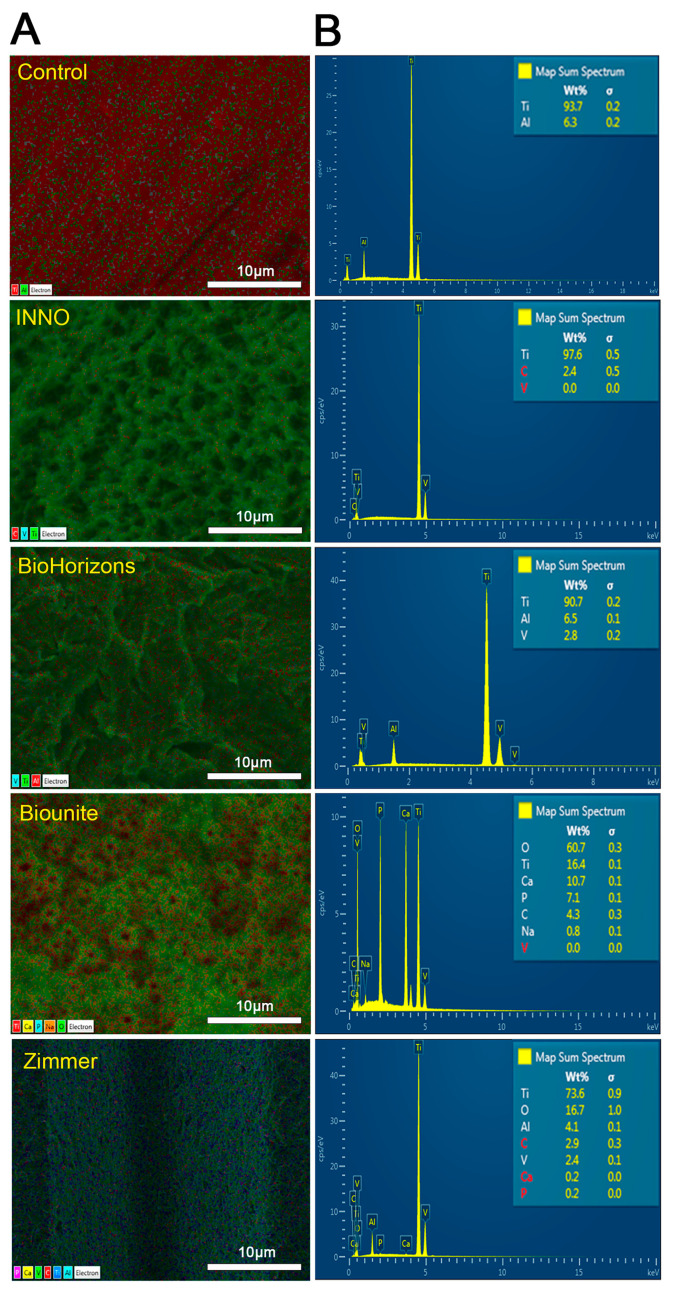
**EDX compositional analysis of the dental implant surfaces.** (**A**) EDX elemental distribution maps on implant surfaces. (**B**) EDX spectra showing the weight percentage values of present elements on the implant surfaces (*n* = 3).

**Figure 5 ijms-24-16686-f005:**
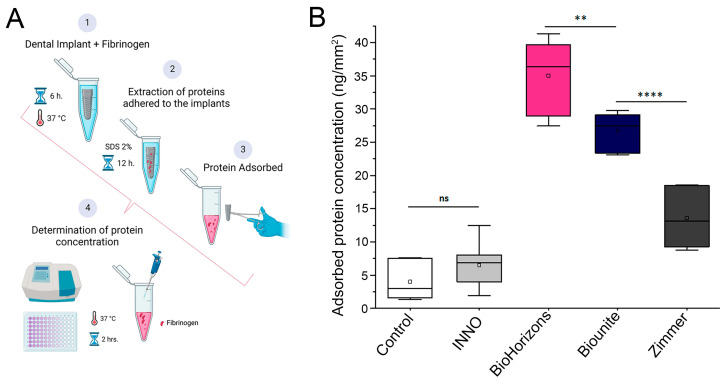
**Amount of fibrinogen protein adsorbed on the implant surfaces.** (**A**) Schematic representation of the experimental procedures associated with protein adsorption assays. (**B**) Following 6 h of incubation, the fibrinogen content was significantly higher for the BioHorizons^TM^ implant compared to the control (*** p* < 0.01, ***** p* < 0.00001), followed by the Biounite^®^ and Zimmer implants. Significantly lower values were found for the INNO implant, which exhibited no significant differences (ns) compared to the control (*p* = 0.75). The graph shows the mean ± SD of three independent experiments (*n* = 3). Values were analyzed with OriginPro 2021 (v.9.8) statistical software using ANOVA followed by the Tukey post-hoc test.

**Figure 6 ijms-24-16686-f006:**
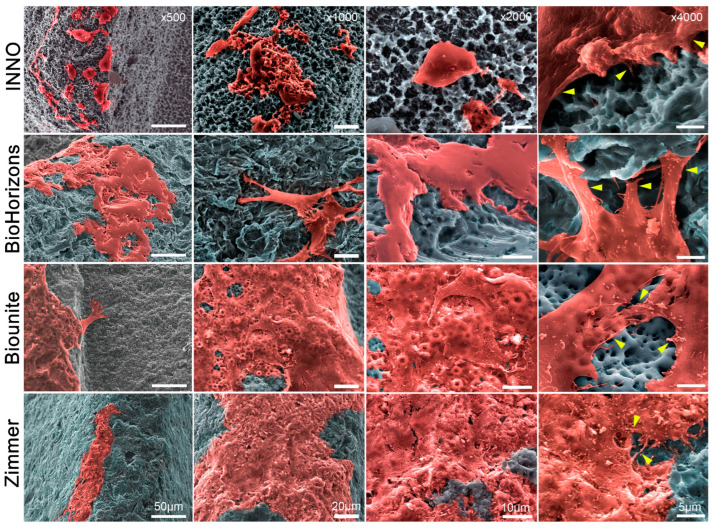
**Analysis of the morphology of mesenchymal stem cells in the implants via SEM.** The GMSCs were incubated for 48 h with the different implants. Subsequently, after fixation and performance of several preanalytical procedures (see the methods section for details), the implants with the cells were subjected to analysis via scanning electron microscopy (SEM). The various images were captured at different magnification levels: 500×, 1000×, 2000×, and 4000×, and display both individual and grouped cells. It is important to highlight the presence of elements, known as filopodia (yellow arrowheads), which are related to the cell adhesion process, as evidenced in the high-magnification images, presented in the fourth column (4000×).

**Figure 7 ijms-24-16686-f007:**
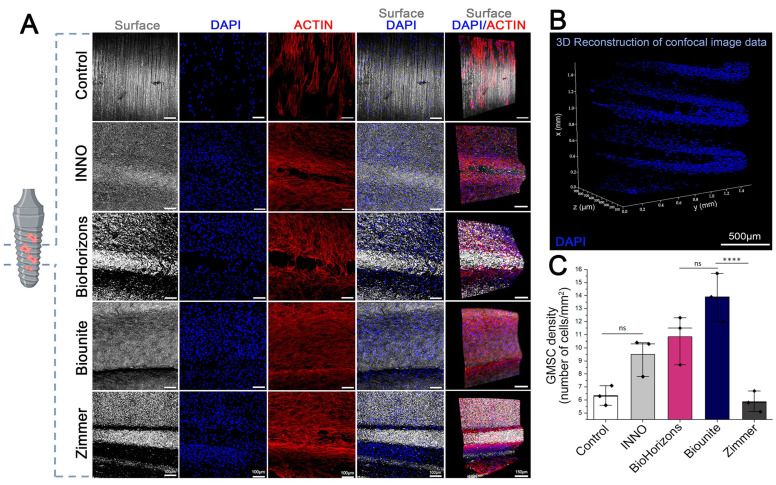
**Representative confocal images of GMSCs at 48 hrs. of contact with implants.** (**A**) Images of nuclei, actin cytoskeleton, and surface obtained via LSCM of the cells attached to the implants at 48 h. The implants were incubated with a mixture of reagents before analysis: Alexa Fluor 555-conjugated phalloidin to visualize actin and DAPI to visualize cell nuclei. The representative images shown in the figure were acquired in the middle zone of the implant. The surface images and 3D reconstruction were obtained using the Argon 488 laser reflection technique. Magnification: 10× and 20×. (**B**) Three-dimensional reconstruction from confocal Z-stacks of the middle area of an implant with cells attached to the surface. (**C**) The number of GMSCs attached per implant has been presented as a graph of cell density. The quantity of adhered cells was significantly higher in the Biounite^®^ and BioHorizons^TM^ implants compared to the control (***** p* < 0.00001) and INNO and Zimmer implants. The latter two did not exhibit significant differences among themselves or with the control. Values normalized to the geometric surface area of each sample plus its standard deviation (*n =* 3 ± SD).

**Table 1 ijms-24-16686-t001:** Roughness measurements of dental implants.

	Control	INNO	BioHorizons^TM^	Biounite^®^	Zimmer
***Sa* (μm)**	**0.28 ± 0.02**	**1.34 ± 0.36**	**1.36 ± 0.04**	**1.13 ± 0.20**	**0.78 ± 0.17**
*Sq* (μm)	0.33 ± 0.03	1.66 ± 0.37	1.59 ± 0.06	1.35 ± 0.24	0.90 ± 0.19
*Sp* (μm)	0.79 ± 0.11	5.45 ± 0.62	2.97 ± 0.05	3.02 ± 0.85	2.22 ± 0.42
*Sv* (μm)	0.77 ± 0.12	4.19 ± 0.56	4.35 ± 0.40	2.92 ± 0.39	1.89 ± 0.17
*Sz* (μm)	1.56 ± 0.17	9.64 ± 1.08	7.32 ± 0.46	5.94 ± 1.01	4.11 ± 0.51

The values were obtained in an area of 10 μm^2^ via analysis with the atomic force microscopy program Gwyddion 2.59. *Sa:* arithmetic mean height; *Sq*: root-mean-square height; *Sp*: maximum height; *Sv*: maximum valley depth; and *Sz*: maximum peak-to-valley distance (*n =* 3 ± SD).

## Data Availability

Data are contained within the article.

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
