# Peer review of "Influence of Topography and Composition of Commercial Titanium Dental Implants on Cell Adhesion of Human Gingiva-Derived Mesenchymal Stem Cells: An In Vitro Study"

_ijms, 2023, doi:10.3390/ijms242316686_

Round 1

Reviewer 1 Report

Comments and Suggestions for Authors

Manuscript ID: ijms-2701972

Authors have studied the influence of the topography and composition of titanium dental implants (available commercially) on protein adsorption and GMSCs adhesion. The study is interesting and meaningful. I have no objections to the plan of experiments, methodology, presentation of results, and discussion - overall, the draft is of a high standard. Nevertheless, I have a few comments and questions for authors that may improve the substantive value of the article:

1) In Table 1 and Figure 4 please add the number of measurement repetitions (n=x).
2)
You stated: ‘’ The presence of these elements in the EDX confirms this premise. It is worth mentioning the differences that these implants present in the amount of titanium (Ti), phosphorus (P), oxygen (O), and calcium (Ca) elements, which is not always reported by the manufacturer.” - Was the composition of the implants obtained in EDX analysis consistent with the composition provided by the manufacturer? And to what extent?
3)
  Line 350 contains an editorial error.
4) In the discussion, please explain more reliably the potential influence of Ca and P ions on GMSCs behavior. This sentence is not sufficient: ‘’Based on this, we believe that the presence of calcium phosphate could be more influential in cell differentiation than in adhesion, a parameter that was not the objective of this study”
5)
 The names in vitro and in vivo should be in italics.
6)
You said: ‘’It is worth noting the hydrophilicity of the Biounite® and BioHorizonsTM implants, which positively influenced the interface between the surface and the proteins, thus improving adsorption.” - Why was not this study conducted for your research? Please provide the experiments or discuss the approximate contact angles for the tested implants.

ikona Zweryfikowane przez spoÅ‚eczność        

Author Response

Dear reviewer,

Thank you very much for your comments. In the attachment, you will find our responses to your requirements.

Reviewer 2 Report

Comments and Suggestions for Authors

This is a good project worthy of being published. This study focuses on investigating the impact of topography and composition of commercial titanium dental implants on the adhesion of human gingiva-derived mesenchymal stem cells (GMSCs). The surface properties of different implants were characterized, and cell adhesion was evaluated after 48 hours. The  findings indicate that the implant with calcium phosphate treatment and regular topography exhibited the highest protein adsorption capacity and density of adherent GMSCs. The surface regularity of the implants was identified as a key factor in the cell adhesion process.

 Some minor concerns: 

Firstly, the introduction appears to be long and could be more focused. Additionally, the first part of the discussion section (384-403) should either be included in the introduction or eliminated entirely.

Furthermore, I noticed that the amount of carbon was not estimated for the EDX technique in the Results section. It would be beneficial to use other techniques such as XPS to detect this important component. Additionally, the Discussion section should begin by analyzing the Null Hypothesis.

Lastly, I recommend discussing the presence of adventitious carbon on the surface

Author Response

(The authors gave the same response as above.)

Reviewer 3 Report

Comments and Suggestions for Authors

Dear Authors,

It was a pleasure to read your article. I believe your paper might be interesting to readers from the clinical field. Your paper is well written and organized.

However, there are some scopes to improve the quality of the manuscript. The reviewer would like to suggest the following revision in the manuscript to make it suitable for publication. 

The aim of this technical note Influence of Topography and Composition of Commercial Titanium Dental Implants on Cell Adhesion of Human Gingiva-derived Mesenchymal Stem Cells: An in Vitro Study" was to characterize the topography and composition of commercial titanium dental implants manufactured with different surface treatments (two sandblasted/acid-etched (SLA) (INNO Implants, Korea; BioHorizonsTM, US) and two calcium phosphate (CaP) treated (Biounite®, Argentina; Zimmer Biomet, Inc., US)) and to investigate their influence on the process of cell adhesion in vitro. 

Minor editing of English language required, Punctuation should be corrected, Standardize text structure and alignment according to guidelines.

Incorrect citation:
"The basic requirements for successful osseointegration were codified by Branemark et al., in the early 1980 [35, 36]". shoud be
"The basic requirements for successful osseointegration were codified by Branemark et al. [35], in the early 1980."
Improve throughout the text.

Results: p value should be written in italics. line 187 - correct table - Line 2 is unreadable.

Material and methods.

Add information about the inclusion and exclusion criteria for the study. p value should be written in italics.

Appendix should be added as a new file, Prepare references according to MDPI guidelines, Reconsider after major revision (control missing in some experiments)

Comments on the Quality of English Language

Minor editing of English language required

Author Response

(The authors gave the same response as above.)
